# Learning disentangled representations with the Wasserstein Autoencoder

## Abstract

Disentangled representation learning has undoubtedly benefited from objective function surgery. However, a delicate balancing act of tuning is still required in order to trade off reconstruction fidelity versus disentanglement. Building on previous successes of penalizing the total correlation in the latent variables, we propose TCWAE (Total Correlation Wasserstein Autoencoder). Working in the WAE paradigm naturally enables the separation of the total-correlation term, thus providing disentanglement control over the learned representation, while offering more flexibility in the choice of reconstruction cost. We propose two variants using different KL estimators and perform extensive quantitative comparisons on data sets with known generative factors, showing competitive results relative to state-of-the-art techniques. We further study the trade off between disentanglement and reconstruction on more-difficult data sets with unknown generative factors, where the flexibility of the WAE paradigm in the reconstruction term improves reconstructions.

## 1 Introduction

Learning representations of data is at the heart of deep learning; the ability to interpret those representations empowers practitioners to improve the performance and robustness of their models (Bengio et al., 2013; van Steenkiste et al., 2019). In the case where the data is underpinned by independent latent generative factors, a good representation should encode information about the data in a semantically meaningful manner with statistically independent latent variables encoding for each factor. Bengio et al. (2013) define a disentangled representation as having the property that a change in one dimension corresponds to a change in one factor of variation, while being relatively invariant to changes in other factors. While many attempts to formalize this concept have been proposed (Higgins et al., 2018; Eastwood & Williams, 2018; Do & Tran, 2019), finding a principled and reproducible approach to assess disentanglement is still an open problem (Locatello et al., 2019).

Recent successful unsupervised learning methods have shown how simply modifying the ELBO objective, either re-weighting the latent regularization terms or directly regularizing the statistical dependencies in the latent, can be effective in learning disentangled representation. Higgins et al. (2017) and Burgess et al. (2018) control the information bottleneck capacity of Variational Autoencoders (VAEs, (Kingma & Welling, 2014; Rezende et al., 2014)) by heavily penalizing the latent regularization term. Chen et al. (2018) perform ELBO surgery to isolate the terms at the origin of disentanglement in $\beta$-VAE, improving the reconstruction-disentanglement trade off. Esmaeili et al. (2018) further improve the reconstruction capacity of $\beta$-TCVAE by introducing structural dependencies both between groups of variables and between variables within each group. Alternatively, directly regularizing the aggregated posterior to the prior with density-free divergences (Zhao et al., 2019) or moments matching (Kumar et al., 2018), or simply penalizing a high Total Correlation (TC, (Watanabe, 1960)) in the latent (Kim & Mnih, 2018) has shown good disentanglement performances.

In fact, information theory has been a fertile ground to tackle representation learning. Achille & Soatto (2018) re-interpret VAEs from an Information Bottleneck view (Tishby et al., 1999), re-phrasing it as a trade off between sufficiency and minimality of the representation, regularizing a pseudo TC between the aggregated posterior and the true conditional posterior. Similarly, Gao et al. (2019) use the principle of total Correlation Explanation (CorEX) (Ver Steeg & Galstyan, 2014) and maximize the mutual information between the observation and a subset of anchor latent points. Maximizing the

mutual information (MI) between the observation and the latent has been broadly used (van den Oord et al., 2018; Hjelm et al., 2019; Bachman et al., 2019; Tschannen et al., 2020), showing encouraging results in representation learning. However, Tschannen et al. (2020) argued that MI maximization alone cannot explain the disentanglement performances of these methods.

Building on the Optimal Transport (OT) problem (Villani, 2008), Tolstikhin et al. (2018) introduced the Wasserstein Autoencoder (WAE), an alternative to VAE for learning generative models. Similarly to VAE, WAE maps the data into a (low-dimensional) latent space while regularizing the averaged encoding distribution. This is in contrast with VAEs where the posterior is regularized at each data point, and allows the encoding distribution to capture significant information about the data while still matching the prior when averaged over the whole data set. Interestingly, by directly regularizing the aggregated posterior, WAE hints at more explicit control on the way the information is encoded, and thus better disentanglement. The reconstruction term of the WAE allows for any cost function on the observation space, opening the door to better suited reconstruction terms, for example when working with continuous RGB data sets where the Euclidean distance or any metric on the observation space can result in more accurate reconstructions of the data.

In this work, following the success of regularizing the TC in disentanglement, we propose to use the Kullback-Leibler (KL) divergence as the latent regularization function in the WAE. We introduce the Total Correlation WAE (TCWAE) with an explicit dependency on the TC of the aggregated posterior. Using two different estimators for the KL terms, we perform extensive comparison with succesful methods on a number of data sets. Our results show that TCWAEs achieve competitive disentanglement performances while improving modelling performance by allowing flexibility in the choice of reconstruction cost.

## 2 IMPORTANCE OF TOTAL CORRELATION IN DISENTANGLEMENT

### 2.1 TOTAL CORRELATION

The TC of a random vector $Z \in \mathcal{Z}$ under $P$ is defined by

$$\mathbf{TC}(Z) \triangleq \sum_{d=1}^{d_Z} H_{p_d}(Z_d) - H_p(Z) \tag{1}$$

where $p_d(z_d)$ is the marginal density over only $z_d$ and $H_p(Z) \triangleq -\mathbb{E}_p \log p(Z)$ is the Shannon differential entropy, which encodes the information contained in $Z$ under $P$. Since

$$\sum_{d=1}^{d_Z} H_{p_d}(Z_d) \leq H_p(Z) \tag{2}$$

with equality when the marginals $Z_d$ are mutually independent, the TC can be interpreted as the loss of information when assuming mutual independence of the $Z_d$; namely, it measures the mutual dependence of the marginals. Thus, in the context of disentanglement learning, we seek a low TC of the aggregated posterior, $p(z) = \int_{\mathcal{X}} p(z|x) \, p(x) \, dx$, which forces the model to encode the data into statistically independent latent codes. High MI between the data and the latent is then obtained when the posterior, $p(z|x)$, manages to capture relevant information from the data.

### 2.2 TOTAL CORRELATION IN ELBO

We consider latent generative models $p_\theta(x) = \int_{\mathcal{Z}} p_\theta(x|z) \, p(z) \, dz$ with prior $p(z)$ and decoder network, $p_\theta(x|z)$, parametrized by $\theta$. VAEs approximate the intractable posterior $p(z|x)$ by introducing an encoding distribution (the encoder), $q_\phi(z|x)$, and learning simultaneously $\theta$ and $\phi$ when optimizing the variational lower bound, or ELBO, defined in Eq. 3:

$$\mathcal{L}_{ELBO}(\theta, \phi) \triangleq \underset{p_{\text{data}}(X)}{\mathbb{E}} \Big[ \underset{q_\phi(Z|X)}{\mathbb{E}} [\log p_\theta(X|Z)] - \mathbf{KL}\Big( q_\phi(Z|X) \parallel p(Z) \Big) \Big] \leq \underset{p_{\text{data}}(X)}{\mathbb{E}} \log p_\theta(X) \tag{3}$$

Following Hoffman & Johnson (2016), we can decompose the KL term in Eq. 3 as:

$$\frac{1}{N_{\text{batch}}} \sum_{n=1}^{N} \mathbf{KL}\Big(q_\phi(Z|x_n) \parallel p(Z)\Big) = \underbrace{\mathbf{KL}\Big(q(Z,N) \parallel q(Z)p(N)\Big)}_{\text{(i) index-code MI}} + \underbrace{\mathbf{KL}\Big(q(Z) \parallel p(Z)\Big)}_{\text{(ii) marginal KL}} \quad (4)$$

where $p(n) = \frac{1}{N}$, $q(z|n) = q(z|x_n)$, $q(z,n) = q(z|n)p(n)$ and $q(z) = \sum_{n=1}^{N} q(z|n)\,p(n)$. (i) refers to the *index-code mutual information* and represents the MI between the data and the latent under the join distribution $q(z,n)$, and (ii) to the *marginal KL* matching the aggregated posterior to the prior. While discussion on the impact of a high index-code MI on disentanglement learning is still open, the marginal KL term plays an important role in disentanglement. Indeed, it pushes the encoder network to match the prior when *averaged*, as opposed to matching the prior for each data point. Combined with a factorized prior $p(z) = \prod_d p_d(z_d)$, as it is often the case, the aggregated posterior is forced to factorize and align with the axis of the prior. More specifically, the marginal KL term in Eq. 4 can be decomposed the as sum of a TC term and a dimensionwise-KL term:

$$\mathbf{KL}\Big(q(Z) \parallel p(Z)\Big) = \mathbf{TC}\Big(q(Z)\Big) + \sum_{d=1}^{d_{\mathcal{Z}}} \mathbf{KL}\Big(q_d(Z_d) \parallel p_d(Z_d)\Big) \quad (5)$$

Thus maximizing the ELBO implicitly minimizes the TC of the aggregated posterior, enforcing the aggregated posterior to disentangle as Higgins et al. (2017) and Burgess et al. (2018) observed when strongly penalizing the KL term in Eq. 3. Chen et al. (2018) leverage the KL decomposition in Eq. 5 by refining the heavy latent penalization to the TC only. However, the index-code MI term in Eq. 4 seems to have little to no role in disentanglement (see ablation study of Chen et al. (2018)), potentially arming the reconstruction performances (Hoffman & Johnson, 2016).

## 3 WAE NATURALLY GOOD AT DISENTANGLING?

In this section we introduce the OT problem and the WAE objective, and discuss the compelling properties of WAEs for representation learning. Mirroring $\beta$-TCVAE decomposition, we derive the TCWAE objective.

### 3.1 WAE

The Kantorovich formulation of the OT between the true-but-unknown data distribution $P_D$ and the model distribution $P_\theta$, for a given cost function $c$, is defined by:

$$\text{OT}_c(P_D, P_\theta) = \inf_{\Gamma \in \mathcal{P}(P_D, P_\theta)} \int_{\mathcal{X} \times \mathcal{X}} c(x, \tilde{x})\, \gamma(x, \tilde{x})\, dx d\tilde{x} \quad (6)$$

where $\mathcal{P}(P_D, P_\theta)$ is the space of all couplings of $P_D$ and $P_\theta$; namely, the space of joint distributions $\Gamma$ on $\mathcal{X} \times \mathcal{X}$ whose densities $\gamma$ have marginals $p_D$ and $p_\theta$. Tolstikhin et al. (2018) derive the WAE objective by restraining this space and relaxing the hard constraint on the marginal using a soft constraint with a Lagrange multiplier (see Appendix A for more details):

$$W_{\mathcal{D},c}(\theta, \phi) \triangleq \mathbb{E}_{p_D(x)} \mathbb{E}_{q_\phi(z|x)} \mathbb{E}_{p_\theta(\tilde{x}|z)} c(x, \tilde{x}) + \lambda\, \mathcal{D}\Big(q(Z) \parallel p(Z)\Big) \quad (7)$$

where $\mathcal{D}$ is any divergence function and $\lambda$ a relaxation parameter. The decoder, $p_\theta(\tilde{x}|z)$, and the encoder, $q_\phi(z|x)$, are optimized simultaneously by dropping the closed-form minimization over the encoder network, with standard stochastic gradient descent methods.

Similarly to the ELBO, objective 7 consists of a reconstruction cost term and a latent regularization term, preventing the latent codes to drift away from the prior. However, WAE explicitly penalizes the aggregate posterior. This motivates, following Section 2.2, the use of WAE in disentanglement learning. Rubenstein et al. (2018) have shown promising disentanglement performances without modifying the objective 7. Another important difference lies in the functional form of the reconstruction cost in the reconstruction term. Indeed, WAE allows for more flexibility in the reconstruction term with any cost function allowed, and in particular, it allows for cost functions better suited to the data at hand and for the use of deterministic decoder networks (Tolstikhin et al., 2018; Frogner et al., 2015). This can potentially result in an improved reconstruction-disentanglement trade off as we empirically find in Sections 4.2 and 4.1.

### 3.2 TCWAE

In this section, for notation simplicity, we drop the explicit dependency of the distributions to their respective parameters.

Following Section 2.2 and Eq. 5, we chose the divergence function, $\mathcal{D}$, in Eq. 7, to be the KL divergence and assume a factorized prior (*e.g.* $p(z) = \mathcal{N}(0_{d_{\mathcal{Z}}}, \mathcal{I}_{d_{\mathcal{Z}}})$), obtaining the same decomposition than in Eq. 5. Re-weighting each term in Eq. 5 with hyper-parameters $\beta$ and $\gamma$, and plugging into Eq. 7, we obtain our TCWAE objective:

$$W_{TC} \triangleq \underset{p(x_n)q(z|x_n)}{\mathbb{E}} \left[ \underset{p(\tilde{x}_n|Z)}{\mathbb{E}} c(x_n, \tilde{x}_n) \right] + \beta\mathbf{KL}\Big(q(Z) \parallel \prod_{d=1}^{d_{\mathcal{Z}}} q_d(Z_d)\Big) + \gamma \sum_{d=1}^{d_{\mathcal{Z}}} \mathbf{KL}\Big(q_d(Z_d) \parallel p_d(Z_d)\Big)$$

$$(8)$$

Given the positivity of the KL divergence, the TCWAE in Eq. 8 is an upper-bound of the WAE objective of Eq. 7 with $\lambda = \min(\beta, \gamma)$.

Eq. 8 can be directly related to the $\beta$-TCVAE objective of Chen et al. (2018):

$$-\mathcal{L}_{\beta-TC} \triangleq \underset{p(x_n)q(z|x_n)}{\mathbb{E}} \Big[ -\log p(x_n|Z) \Big] + \beta\mathbf{KL}\Big(q(Z) \parallel \prod_{d=1}^{d_{\mathcal{Z}}} q_d(Z_d)\Big) + \gamma \sum_{d=1}^{d_{\mathcal{Z}}} \mathbf{KL}\Big(q_d(Z_d) \parallel p_d(Z_d)\Big)$$

$$+ \alpha I_{\mathsf{q}}\Big(q(Z,N); q(Z)p(N)\Big) \tag{9}$$

As already mentioned, the main differences are the absence of index-code MI and a different reconstruction cost function. Setting $\alpha = 0$ in Eq. 9 makes the two latent regularizations match but breaks the inequality in Eq. 3. Matching the two reconstruction terms would be possible if we could find a ground cost function $c$ such that $\mathbb{E}_{p(\tilde{x}_n|Z)}c(x_n, \tilde{x}_n) = -\log p(x_n|Z)$.

### 3.3 ESTIMATORS

While being grounded and motivated by information theory and earlier works on disentanglement, using the KL as the latent divergence function, as opposed to other sampled-based divergences (Tolstikhin et al., 2018; Patrini et al., 2018), presents its own challenges. Indeed, the KL terms are intractable, and especially, we need estimators to approximate the entropy terms. We propose to use two estimators, one based on importance weight-sampling Chen et al. (2018), the other on adversarial estimation using the denisty-ratio trick (Kim & Mnih, 2018).

#### TCWAE-MWS

Chen et al. (2018) propose to estimate the intractable terms $\mathbb{E}_q \log q(Z)$ and $\mathbb{E}_{q_d} \log q_d(Z)$ in the KL terms of Eq. 8 with Minibatch-Weighted Sampling (MWS). Considering a batch of observation $\{x_1, \ldots x_{N_{\text{batch}}}\}$, they sample the latent codes $z_i \sim q(z|x_i)$ and compute:

$$\underset{q(z)}{\mathbb{E}} \log q(z) \approx \frac{1}{N_{\text{batch}}} \sum_{i=1}^{N_{\text{batch}}} \log \frac{1}{N \times N_{\text{batch}}} \sum_{j=1}^{N_{\text{batch}}} q(z_i|x_j) \tag{10}$$

This estimator, while being easily computed from samples, is a biased estimator of $\mathbb{E}_q \log q(Z)$. Chen et al. (2018) also proposed an unbiased version, the Minibatch-Stratified Sampling (MSS). However, they found that it did not result in improved performances, and thus, as Chen et al. (2018), we chose to use the simpler MWS estimator. We call the resulting algorithm the TCWAE-MWS. Other sampled-based estimators of the entropy or the KL divergence have been proposed (Rubenstein et al., 2019; Esmaeili et al., 2018). However, we choose the solution of Chen et al. (2018) for 1) its simplicity and 2) the similarities between the TCWAE and $\beta$-TCVAE objectives.

#### TCWAE-GAN

A different approach, similar in spirit to the WAE-GAN originally proposed by Tolstikhin et al. (2018), is based on adversarial-training. While Tolstikhin et al. (2018) use the adversarial training to approximate the JS divergence, Kim & Mnih (2018) use the density-ratio trick and adversarial

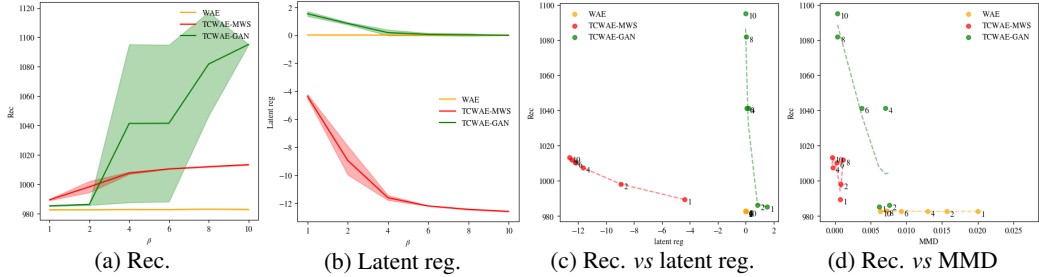

Figure 1: Reconstruction and latent regularization terms as functions of $\beta$ for the NoisydSprites data set. (a): reconstruction error. (b): latent regularization term (MMD for WAE, KL for TCWAE). (c): reconstruction error against latent regularization. (d): reconstruction error against MMD. Shaded regions show $\pm$ one standard deviation.

training to estimate the intractable terms in Eq. 8. The the density-ratio trick (Nguyen et al., 2008; Sugiyama et al., 2011) estimates the KL divergence as:

$$\mathbf{KL}\Big(q(z) \parallel \prod_{d=1}^{d_{\mathcal{Z}}} q_d(z_d)\Big) \approx \mathop{\mathbb{E}}_{q(z)} \log \frac{D(z)}{1 - D(z)} \tag{11}$$

where $D$ plays the same role than the discriminator in GANs and ouputs an estimate of the probability that $z$ is sampled from $q(z)$ and not from $\prod_{d=1}^{d_{\mathcal{Z}}} q_d(z_d)$. Given that we can easily sample from $q(z)$, we can use Monte-Carlo sampling to estimate the expectation in Eq. 11. The discriminator $D$ is adversarially trained alongside the decoder and encoder networks. We call this adversarial version the TCWAE-GAN.

## 4 EXPERIMENTS

We perform a series of quantitative and qualitative experiments, starting with an ablation study on the impact of using different latent regularization functions in WAEs followed by a quantitative comparison of the disentanglement performances of our methods with existing ones on toy data sets before moving to qualitative assessment of our method on more challenging data sets. Details of the data sets, the experimental setup as well as the networks architectures are given in Appendix B. In all the experiments we fix the ground-cost function of the WAE-based methods to be the square Euclidean distance: $c(x, y) = \|x - y\|_{L_2}^2$.

### 4.1 QUANTITATIVE ANALYSIS: DISENTANGLEMENT ON TOY DATA SETS

**Ablation study of the latent divergence function** We compare the impact of the different latent regularization functions in WAE-MMD (Tolstikhin et al., 2018), TCWAE-MWS and TCWAE-GAN. We take $\beta = \gamma$ in the TCWAE objectives isolating the impact of the different latent divergence functions used in the TCWAE and the original WAE. We train the methods with $\beta \in \{1, 2, 4, 6, 8, 10\}$, and report the results Figure 1 in the case of the NoisydSprites data set (Locatello et al., 2019). As expected, the higher the penalization on the latent regularization (high $\beta$), the poorer the reconstructions. We can see that the trade off between reconstruction and latent regularization is more sensible for TCWAE-GAN, where a relatively modest improvement in latent regularization results in an important deterioration of reconstruction performances while TCWAE-MWS is less sensible. This is better illustrated in Figure 1c with a much higher slope for TCWAE-GAN than for TCWAE-MWS. WAE seems to be relatively little impacted by the latent penalization weight. We note in Figure1b the bias of the MWS estimator (Chen et al., 2018). Finally, we plot the reconstruction versus the MMD between the aggregated posterior and the prior for all the models in Figure (1d). Interestingly, TCWAEs actually achieved a lower MMD (left part of the plot) even if they are not being trained with that regularization function. However, as expected given that the TCWAE do not optimized the reconstruction-MMD trade off, the WAE achieved a better reconstruction (bottom part of the plot).

Table 1: Reconstruction and disentanglement scores ($\pm$ one standard deviation).

| Method | MSE | MIG | factorVAE | SAP |
|---|---|---|---|---|
| TCWAE MWS ($\beta = 6$) | $34.95 \pm 0.90$ | $\mathbf{0.323 \pm 0.04}$ | $0.77 \pm 0.01$ | $0.072 \pm 0.004$ |
| TCWAE GAN ($\beta = 10$) | $11.39 \pm 0.28$ | $0.181 \pm 0.01$ | $0.76 \pm 0.03$ | $0.074 \pm 0.003$ |
| $\beta$-TCVAE ($\beta = 6$) | $14.30 \pm 2.43$ | $0.235 \pm 0.03$ | $\mathbf{0.81 \pm 0.03}$ | $0.070 \pm 0.006$ |
| FactorVAE ($\gamma = 10$) | $\mathbf{8.17 \pm 0.86}$ | $0.24 \pm 0.06$ | $0.78 \pm 0.03$ | $\mathbf{0.077 \pm 0.011}$ |

(a) dSprites

| Method | MSE | MIG | factorVAE | SAP |
|---|---|---|---|---|
| WAE ($\lambda = 2$) | $982.51 \pm .20$ | $0.019 \pm .00$ | $0.40 \pm .09$ | $0.011 \pm .005$ |
| TCWAE MWS ($\beta = 2$) | $998.17 \pm 3.82$ | $\mathbf{0.118 \pm .08}$ | $0.57 \pm .07$ | $0.011 \pm .005$ |
| TCWAE GAN ($\beta = 4$) | $\mathbf{986.77 \pm .48}$ | $0.055 \pm .03$ | $\mathbf{0.58 \pm .04}$ | $0.017 \pm .005$ |
| $\beta$-TCVAE ($\beta = 8$) | $998.67 \pm 3.71$ | $0.101 \pm .06$ | $0.53 \pm .11$ | $0.015 \pm .007$ |
| FactorVAE ($\gamma = 25$) | $988.10 \pm .81$ | $0.066 \pm .03$ | $0.52 \pm .07$ | $\mathbf{0.019 \pm .008}$ |

(b) NoisydSprites

| Method | MSE | MIG | factorVAE | SAP |
|---|---|---|---|---|
| WAE ($\lambda = 6$) | $24.40 \pm .43$ | $0.014 \pm .01$ | $0.41 \pm .04$ | $0.010 \pm .004$ |
| TCWAE MWS ($\beta = 2$) | $39.53 \pm .24$ | $\mathbf{0.322 \pm .00}$ | $\mathbf{0.73 \pm .01}$ | $\mathbf{0.067 \pm .001}$ |
| TCWAE GAN ($\beta = 8$) | $33.57 \pm .57$ | $0.158 \pm .02$ | $0.67 \pm .04$ | $0.039 \pm .009$ |
| $\beta$-TCVAE ($\beta = 6$) | $43.64 \pm .28$ | $0.261 \pm .11$ | $0.67 \pm .14$ | $0.053 \pm .020$ |
| FactorVAE ($\gamma = 25$) | $\mathbf{33.23 \pm .53}$ | $0.256 \pm .07$ | $0.69 \pm .09$ | $0.066 \pm .013$ |

(c) ScreamdSprites

| Method | MSE | MIG | factorVAE | SAP |
|---|---|---|---|---|
| WAE ($\lambda = 2$) | $3.85 \pm .0.03$ | $0.010 \pm .000$ | $0.38 \pm .02$ | $0.008 \pm .004$ |
| TCWAE MWS ($\beta = 2$) | $11.48 \pm .26$ | $0.029 \pm .003$ | $0.44 \pm .03$ | $\mathbf{0.017 \pm .002}$ |
| TCWAE GAN ($\beta = 2$) | $\mathbf{6.87 \pm .10}$ | $0.030 \pm .007$ | $0.46 \pm .02$ | $0.015 \pm .001$ |
| $\beta$-TCVAE ($\beta = 4$) | $10.34 \pm .06$ | $0.030 \pm .001$ | $0.46 \pm .02$ | $0.016 \pm .001$ |
| FactorVAE ($\gamma = 100$) | $8.60 \pm .15$ | $\mathbf{0.038 \pm .00}$ | $\mathbf{0.47 \pm .02}$ | $0.015 \pm .003$ |

(d) smallNORB

**Disentanglement performances**   We compare our methods with $\beta$-TCVAE (Chen et al., 2018), FactorVAE (Kim & Mnih, 2018) and the original WAE-MMD (Tolstikhin et al., 2018) on the dSprites (Matthey et al., 2017), NoisydSprites (Locatello et al., 2019), ScreamdSprites (Locatello et al., 2019) and smallNORB (LeCun et al., 2004) data sets whose ground-truth generative-factors are known and given in Table 3, Appendix B.1. We use three different disentanglement metrics to assess the disentanglement performances: the Mutual Information Gap (MIG, Chen et al. (2018)), the factorVAE metric (Kim & Mnih, 2018) and the Separated Attribute Predictability score (SAP, Kumar et al. (2018)). We follow Locatello et al. (2019) for the implementation of these metrics. We use the Mean Square Error (MSE) of the reconstructions to assess the reconstruction performances of the methods. For each model, we use 6 different values for each parameter, resulting in thirty-six different models for TCWAEs, and six for the remaining methods (see Appendix B.1 for more details).

Mirroring the benchmark methods, we first tune $\gamma$ in the TCWAEs, regularizing the dimensionwise-KL, subsequently focusing on the role of the TC term in the disentanglement performances. The heat maps of the different scores for each method and data set are given Figures 5, 6, 7 and 8 in Appendix C. As expected, while $\beta$ controls the trade off between reconstruction and disentanglement, $\gamma$ affects the range achievable when tuning $\beta$. Especially, for $\gamma > 1$, we can see Figures5,6, 7 and 8 that better disentanglement is obtained without much deterioration in reconstruction.

Table 1 reports the results, averaged over 5 random runs, for the four different data sets. For each method, we report the best $\beta$ taken to be the one achieving an overall best ranking on the four different metrics (see Appendix **??** for details). Note that the performances of WAE on the dSprites data set, both in term of reconstruction and disentanglement where significantly worse and meaningless, thus, in order to avoid unfair extra tuning of the parameters, we chose not to include them. TCWAEs achieve competitive performances across all the data sets, with top scores in several metrics. Especially, the square Euclidean distance seems to improve the trade off and perform better than the cross-entropy with color images (NoisydSprites, ScreamdSprites) but less so with black and white images (dSprites). See Appendix C for more results on the different data sets.

As a sanity check, we plot Figure 2 the latent traversals of the different methods on the smallNORB data set. More specifically, we encode one observation and traverse the latent dimensions one at the time (rows) and reconstruct the resulting latent traversals (columns). Visually, all methods, with the exception of WAE, learn to disentangle, capturing four different factors in line with the ground-truth generative factors. Models reconstructions and samples for the different data sets are given in Appendix C.

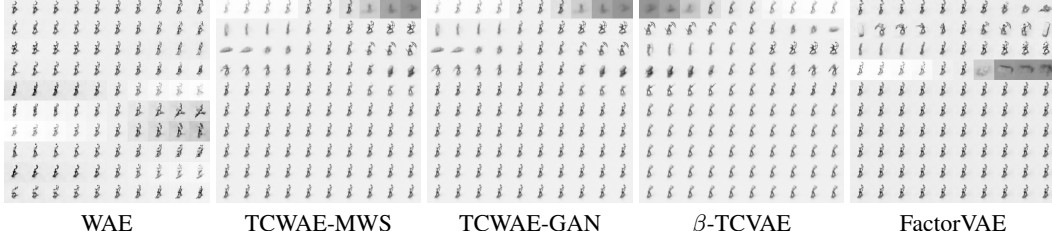

| WAE | TCWAE-MWS | TCWAE-GAN | $\beta$-TCVAE | FactorVAE |

Figure 2: Latent traversals for each model on smallNORB. The parameters are the same than the ones reported in Tables 1 and 7. Each row $i$ corresponds to the traversal of the latent $z_i$ while the columns correspond to a step in the that traversal. The rows are order by increasing $\mathbf{KL}\Big(1/N_{test} \sum_{testset} q(z_i|x) \parallel p(z_i)\Big)$ and the traversal range is $[-2, 2]$.

Finally, we visualise the reconstruction-disentanglement trade off by plotting the different disentanglement metrics against the MSE in Figure 3. As expected, when the TC regularization weight is increased, the reconstruction deteriorates while the disentanglement improves up to a certain point. Then, when too much penalization is put on the TC term, the poor quality of the reconstructions prevents any disentanglement in the generative factors. Reflecting the results of Table 1, TCWAE-MWS seems to perform better (top-left corner represents better reconstruction and disentanglement). TCWAE-GAN presents better reconstruction but slightly lower disentanglement performances (bottom left corner).

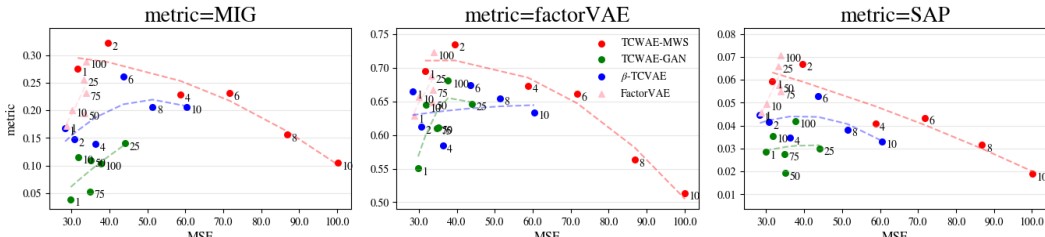

Figure 3: Disentanglement versus reconstruction on the ScreamdSprites data set. Annotations at each point are values of $\beta$. Points with low reconstruction error and high scores (top-left corner) represent better models.

## 4.2 QUALITATIVE ANALYSIS: DISENTANGLEMENT ON REAL-WORLD DATA SETS

We train our methods on 3Dchairs (Aubry et al., 2014) and CelebA (Liu et al., 2015) whose generative factors are not known and qualitatively find that TCWAEs achieve good disentanglement. Figure 4 shows the latent traversals of four different factors learned by the TCWAEs, while Figures 16 and 18 in Appendix D show the models reconstructions and samples. Visually, TCWAEs manage to capture different generative factors while retaining good reconstructions and samples. This confirms our intuition that the flexibility offered in the construction of the reconstruction term, mainly the possibility to chose the reconstruction cost function and use deterministic decoders, improves the reconstruction-disentanglement trade off. In order to assess the quality of the reconstructions, we compute the MSE of the reconstructions and the FID scores (Heusel et al., 2017) of the reconstructions and samples. Results are reported in Table 2. TCWAEs indeed beat their VAEs counterparts in both data sets. It is worth noting that, while the performances of FactorVAE in Table 2 seem good, the

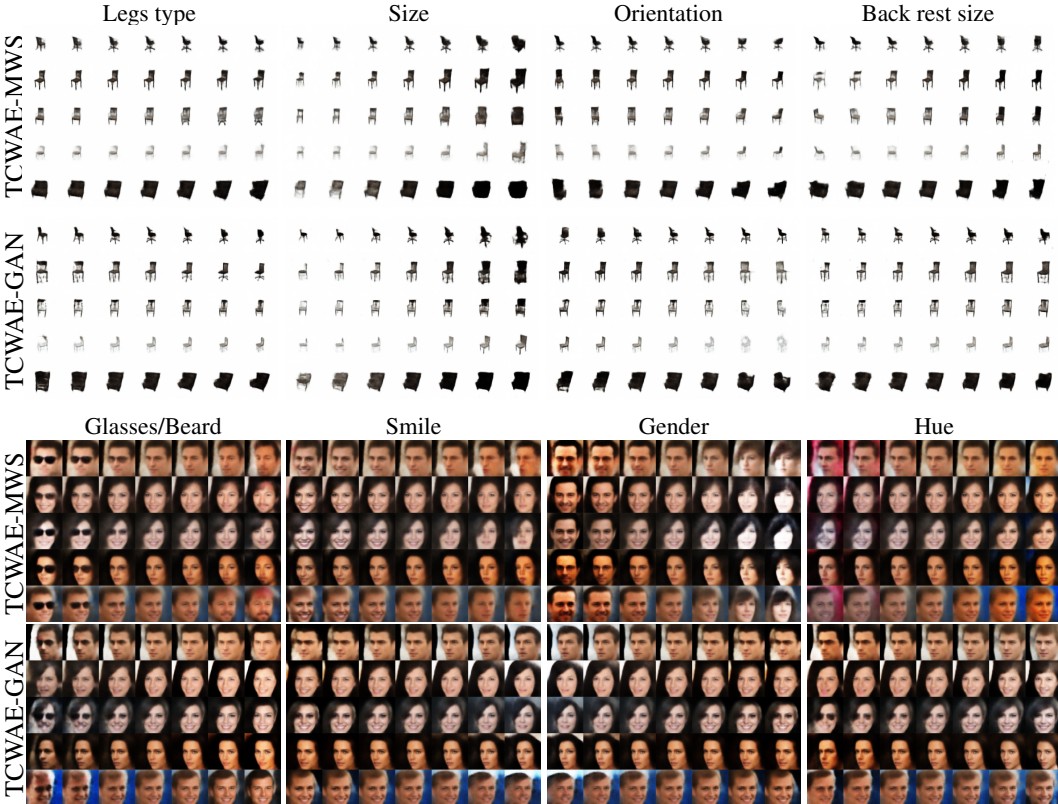

Figure 4: Latent traversals for TCWAE-MWS and TCWAE-GAN. Each line corresponds to one input data point while each subplot corresponds to one latent factor. We vary evenly the encoded latent codes in the interval $[-4, 4]$.

Table 2: MSE and FID scores for the different data sets. Details of the methodology is given in Appendix B

| | 3D chairs | | | CelebA | | |
|---|---|---|---|---|---|---|
| Method | MSE | Rec. | Samples | MSE | Rec. | Samples |
| TCWAE-MWS | $45.8 \pm 4.72$ | 1.227 | 1.821 | $147.5 \pm 33.58$ | 1.204 | 1.264 |
| TCWAE-GAN | $29.8 \pm 3.46$ | 0.518 | 0.362 | $129.8 \pm 34.45$ | 1.003 | 0.975 |
| $\beta$-TCVAE | $43.0 \pm 4.85$ | 1.346 | 1.845 | $180.8 \pm 51.1$ | 1.360 | 1.411 |
| FactorVAE | $42.1 \pm 7.58$ | 0.895 | 0.684 | $201.4 \pm 51.84$ | 1.017 | 0.982 |

inspection of the reconstructions and samples in Appendix D shows that FactorVAE in fact struggle to generalize and to learn a smooth latent manifold.

## 5 CONCLUSION

Leveraging the surgery of the KL regularization term of the ELBO objective, we design a new disentanglement method based on the WAE objective whose latent divergence function is taken to be the KL divergence between the aggregated posterior and the prior. The WAE framework naturally enables the latent regularization to depend explicitly on the TC of the aggregated posterior, quantity previously associated with disentanglement. Using two different estimators of the KL terms, we show that our methods achieve competitive disentanglement on toy data sets. Moreover, the flexibility in the choice of the reconstruction cost function offered by the WAE framework makes our method more compelling when working with more challenging data sets.

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

## A    WAE DERIVATION

We recall the Kantorovich formulation of the OT between the true-but-unknown data distribution $P_D$ and the model distribution $P_\theta$, with given cost function $c$:

$$\text{OT}_c(P_D, P_\theta) = \inf_{\Gamma \in \mathcal{P}(P_D, P_\theta)} \int_{\mathcal{X} \times \mathcal{X}} c(x, \tilde{x}) \, \gamma(x, \tilde{x}) \, dx \, d\tilde{x} \tag{12}$$

where $\mathcal{P}(P_D, P_\theta)$ is the space of all couplings of $P_D$ and $P_\theta$:

$$\mathcal{P}(P_D, P_\theta) = \left\{ \Gamma \,\Big|\, \int_{\mathcal{X}} \gamma(x, \tilde{x}) \, d\tilde{x} = p_D(x), \int_{\mathcal{X}} \gamma(x, \tilde{x}) \, dx = p_\theta(\tilde{x}) \right\} \tag{13}$$

Tolstikhin et al. (2018) first restrain the space of couplings to the joint distributions of the form:

$$\gamma(x, \tilde{x}) = \int_{\mathcal{Z}} p_\theta(\tilde{x}|z) \, q(z|x) \, p_D(x) \, dz \tag{14}$$

where $q(z|x)$, for $x \in \mathcal{X}$, plays the same role as the variational distribution in variational inference.

While the marginal constraint on $x$ (first constraint in Eq. 13) in Eq. 14 is satisfied by construction, the second marginal constraint (that over $x$ giving $p_\theta$ in in Eq. 13) is not guaranteed. A sufficient condition is to have for all $z \in \mathcal{Z}$:

$$\int_{\mathcal{X}} q(z|x) \, p_D(x) \, dx = p(z) \tag{15}$$

Secondly, Tolstikhin et al. (2018) relax the constraint in Eq. 15 using a soft constraint with a Lagrange multiplier:

$$\widehat{W}_c(P_D, P_\theta) = \inf_{q(Z|X)} \left[ \int_{\mathcal{X} \times \mathcal{X}} c(x, \tilde{x}) \, \gamma(x, \tilde{x}) \, dx \, d\tilde{x} + \lambda \, \mathcal{D}\Big( q(Z) \, \| \, p(Z) \Big) \right] \tag{16}$$

where $\mathcal{D}$ is any divergence function, $\lambda$ a relaxation parameter, $\gamma$ is defined in Eq. 14 and $q(Z)$ is the aggregated posterior as define in Section 2. Finally, they drop the closed-form minimization over the variational distribution $q(z|x)$, to obtain the WAE objective, as defined in Section 3.1:

$$\begin{aligned}
W_{\mathcal{D},c}(\theta, \phi) &\triangleq \mathop{\mathbb{E}}_{p_D(X)} \mathop{\mathbb{E}}_{q_\phi(z|x)} \mathop{\mathbb{E}}_{p_\theta(\tilde{x}|z)} c(x, \tilde{x}) + \lambda \, \mathcal{D}\Big( q(Z) \, \| \, p(Z) \Big) \\
&\approx \mathop{\mathbb{E}}_{p(x_n)} \mathop{\mathbb{E}}_{q_\phi(z|x_n)} \mathop{\mathbb{E}}_{p_\theta(\tilde{x}_n|z)} c(x, \tilde{x}_n) + \lambda \, \mathcal{D}\Big( q(Z) \, \| \, p(Z) \Big)
\end{aligned} \tag{17}$$

## B    IMPLEMENTATION DETAILS

### B.1    EXPERIMENTAL SETUP

We train and compare our methods on four different data sets, two with known ground-truth generative factors (see Table 3): dSprites (Matthey et al., 2017) with 737,280 binary, $64 \times 64$ images and smallNORB (LeCun et al., 2004) with 48,600 greyscale, $64 \times 64$ images; and two with unknown ground-truth generative factors: 3Dchairs (Aubry et al., 2014) with 86,366 RGB, $64 \times 64$ images and CelebA (Liu et al., 2015) with 202,599 RGB $64 \times 64$ images.

Table 3: Ground-truth generative-factors of the dSprites and smallNORB data sets.

| data set | Generative factors (number of different values) |
|---|---|
| dSprites and variations | Shape (3), Orientation (40), Position X (32), Position Y (32) |
| smallNORB | categories (5), lightings (6), elevations (9), azimuths (18) |

We use a batch size of 64 in Section 4.2, while in the main experiments of Section 4.1, we take a batch size of 100. In the ablation study of Section 4.1, we use a bigger batch size of 256 in order to reduce the impact of the bias of the MWS estimator (Chen et al. (2018) however show that

Table 4: Hyper parameters values ranges used in the different Sections.

| Method | Section 4.2 | Section 4.1 |
|---|---|---|
| TCWAE-MWS | $\{1, 2, 4, 6, 8, 10\}^2$ | $\{1, 2, 5, 10, 15, 20\}^2$ |
| TCWAE-GAN | $\{1, 2, 4, 6, 8, 10\}^2$ | $\{1, 2, 5, 10, 20, 50\}^2$ |
| $\beta$-TCVAE | $\{1, 2, 4, 6, 8, 10\}$ | $\{1, 2, 5, 10, 15, 20\}$ |
| FactorVAE | $\{1, 10, 25, 50, 75, 100\}$ | $\{1, 2, 5, 10, 20, 50\}$ |

there is very little impact on the performance of the MWS when using smaller batch size). For all experiments, we use the Adam optimizer (Kingma & Ba, 2015) with a learning rate of 0.0005, beta1 of 0.9, beta2 of 0.999 and epsilon of 0.0008 and train for 300,000 iterations. For all the data sets of Section 4.1, we take the latent dimension $d_{\mathcal{Z}} = 10$, while we use $d_{\mathcal{Z}} = 16$ for 3Dchairs and $d_{\mathcal{Z}} = 32$ for CelebA. We use Gaussian encoders with diagonal covariance matrix in all the models and deterministic decoder networks when possible (WAE-based methods). We follow Locatello et al. (2019) for the architectures in all the experiments expect for CelebA where we follow Tolstikhin et al. (2018) (details of the networks architectures given Section B.2). We use a (positive) mixture of Inverse MultiQuadratic (IMQ) kernels and the associated reproductive Hilbert space to compute the MMD when it is needed (WAE and ablation study of Section 4.1).

The different parameter values used for each experiment are given Table 4. $\gamma$ is chosen such that the resulting method achieves the best score $s$, when averaging over all the $\beta$ values, where the score is defined as the sum of the ranking on each individual metric: $s = r_{MSE} + \sum_{metric} r_{metric}$ where $r_{MSE}$ designed the ranking of the MSE (lower is better) and $r_{metric}$, for $metric$ in {MIG, FactorVAE, SAP}, is the ranking of the disentanglement performances as measured by the given metric (higher is better). $\beta$ is then chosen such that the resulting method, with the previously found $\gamma$, achieves the best overall score $s$ defined above. In Section 4.1, we use a validation run to select the parameters values and report the MSE and FID scores on a test run. MSE are computed on a test set of size 10,000 with batch size of 1,000, while we follow Heusel et al. (2017) for the FID implementation: we first compute the activation statistics of the features maps on the full test set for both the reconstruction, respectively samples, and the true observations. We then compute the Frechet distance between two Gaussian with the computed statistics.

## B.2 MODELS ARCHITECTURES

The Gaussian encoder networks, $q_\phi(z|x)$ and decoder network, $p_\theta(x|z)$, are parametrized by neural networks as follow:

$$p_\theta(x|z) = \begin{cases} \delta_{\boldsymbol{f_\theta(z)}} & \text{if WAE based method,} \\ \mathcal{N}\big(\boldsymbol{\mu_\theta}(z), \boldsymbol{\sigma_\theta^2}(z)\big) & \text{otherwise.} \end{cases}$$

$$q_\phi(z|x) = \mathcal{N}\big(\boldsymbol{\mu_\phi}(x), \boldsymbol{\sigma_\phi^2}(x)\big)$$

where $\boldsymbol{f_\theta}, \boldsymbol{\mu_\theta}, \boldsymbol{\sigma_\theta^2}, \boldsymbol{\mu_\phi}$ and $\boldsymbol{\sigma_\phi^2}$ are the outputs of convolutional neural networks. All the experiments use the architectures of Locatello et al. (2019) except for CelebA where we use the architecture inspired by Tolstikhin et al. (2018). The details for the architectures are given Table 5.

All the discriminator networks, $D$, are fully connected networks and share the same architecture given Table 5. The optimisation setup for the discriminator is given Table 6.

Table 5: Networks architectures

| Encoder | Decoder | Discriminator |
|---|---|---|
| Input: $64 \times 64 \times$ c | Input: $d_{\mathcal{Z}}$ | Input: $d_{\mathcal{Z}}$ |
| CONV. $4 \times 4 \times 32$ stride 2 ReLU | FC 256 ReLU | FC 1000 ReLU |
| CONV. $4 \times 4 \times 32$ stride 2 ReLU | FC $4 \times 4 \times 64$ ReLU | FC 1000 ReLU |
| CONV. $4 \times 4 \times 64$ stride 2 ReLU | CONV. $4 \times 4 \times 64$ stride 2 ReLU | FC 1000 ReLU |
| CONV. $4 \times 4 \times 64$ stride 2 ReLU | CONV. $4 \times 4 \times 32$ stride 2 ReLU | FC 1000 ReLU |
| FC 256 Relu | CONV. $4 \times 4 \times 32$ stride 2 ReLU | FC 1000 ReLU |
| FC $2 \times d_{\mathcal{Z}}$ | CONV. $4 \times 4 \times c$ stride 2 | FC 1000 ReLU |
| | | FC 2 |

(a) Locatello et al. (2019) architectures

| Encoder | Decoder | Discriminator |
|---|---|---|
| Input: $64 \times 64 \times c$ | Input: $d_{\mathcal{Z}}$ | Input: $d_{\mathcal{Z}}$ |
| CONV. $4 \times 4 \times 32$ stride 2 BN ReLU | FC $8 \times 8 \times 256$ BN ReLU | FC 1000 ReLU |
| CONV. $4 \times 4 \times 64$ stride 2 BN ReLU | CONV. $4 \times 4 \times 128$ stride 2 BN ReLU | FC 1000 ReLU |
| CONV. $4 \times 4 \times 128$ stride 2 BN ReLU | CONV. $4 \times 4 \times 64$ stride 2 BN ReLU | FC 1000 ReLU |
| CONV. $4 \times 4 \times 256$ stride 2 BN ReLU | CONV. $4 \times 4 \times 32$stride 2 BN Relu | FC 1000 ReLU |
| FC $2 \times d_{\mathcal{Z}}$ | CONV. $4 \times 4 \times c$ | FC 1000 ReLU |
| | | FC 1000 ReLU |
| | | FC 2 |

(b) CelebA networks architectures

Table 6: FactorVAE discriminator setup

| Parameter | Value |
|---|---|
| Learning rate | $1e^{-4}$ (Section 4.1) / $1e^{-5}$ (Section 4.2) |
| beta 1 | 0.5 |
| beta 2 | 0.9 |
| epsilon | 1e-08 |

# C QUANTITATIVE EXPERIMENTS

HYPER PARAMETER TUNING

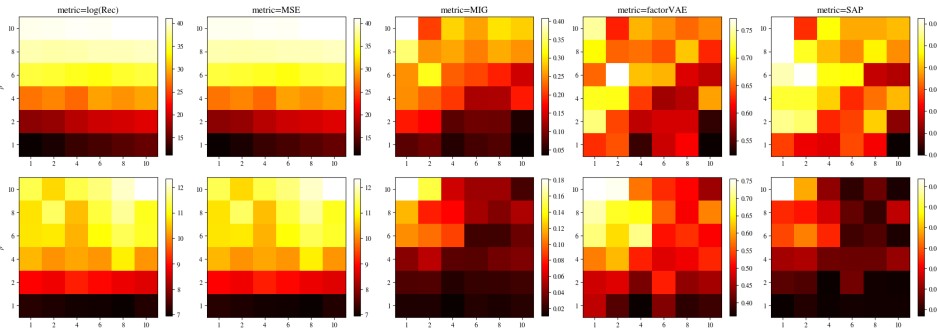

Figure 5: Heat maps for the different scores on dSprites.

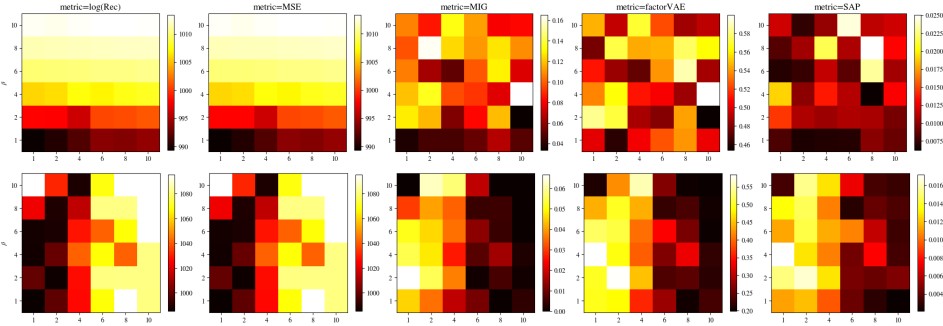

Figure 6: Heat maps for the different scores on NoisydSprites.

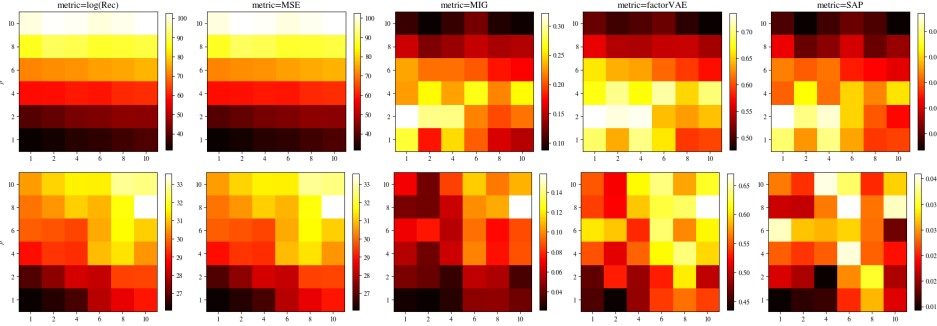

Figure 7: Heat maps for the different scores on ScreamdSprites.

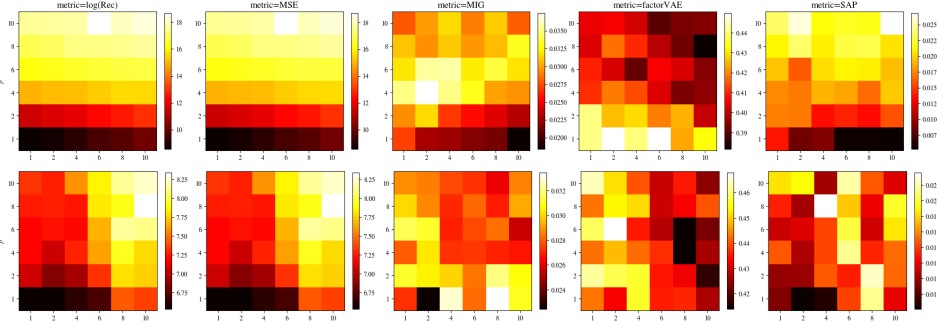

Figure 8: Heat maps for the different scores on smallNORB.

Table 7: $\gamma$ values for methods for each data set.

| Method | dSprites | NoisydSprites | ScreamdSprites | smallNORB |
|---|---|---|---|---|
| TCWAE MWS | 2 | 2 | 1 | 1 |
| TCWAE GAN | 1 | 1 | 10 | 2 |

DISENTANGLEMENT SCORES *vs* $\beta$

For each method, we plot the distribution (over five random runs) of the different metrics for different $\beta$ values.

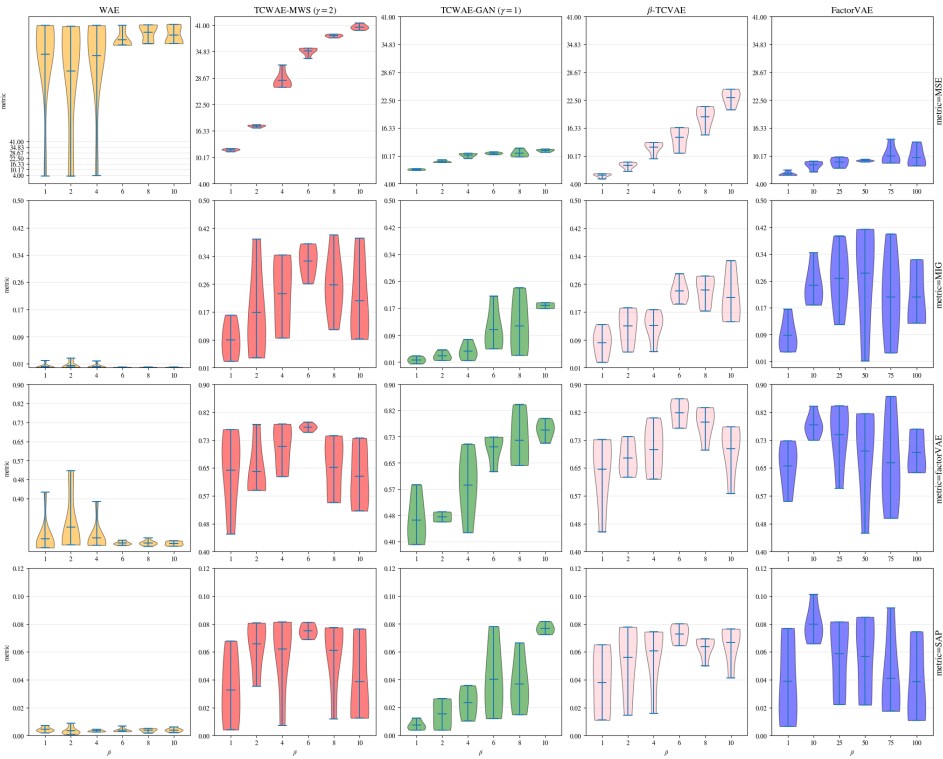

Figure 9: Violin plots of the different scores versus $\gamma$ on dSprites.

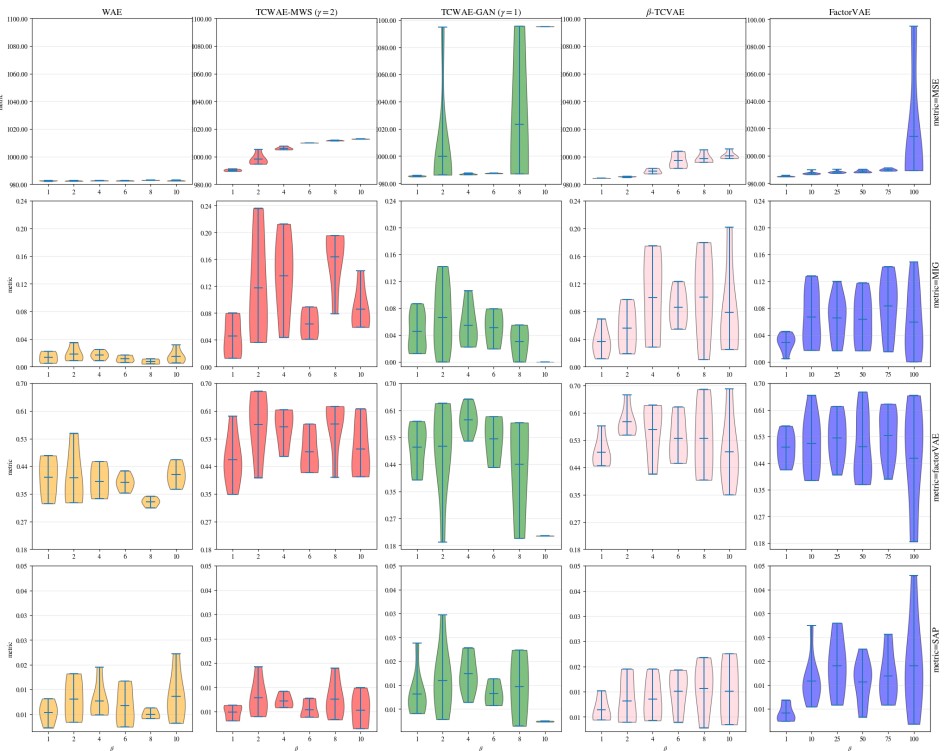

Figure 10: Violin plots of the different scores versus $\gamma$ on NoisydSprites.

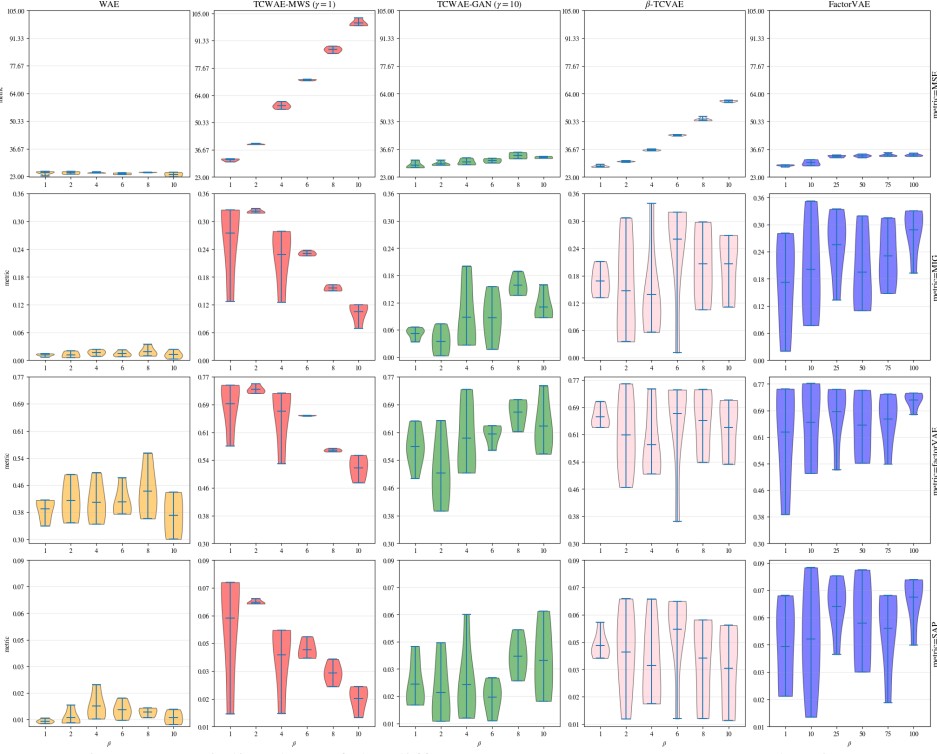

Figure 11: Violin plots of the different scores versus $\gamma$ on ScreamdSprites.

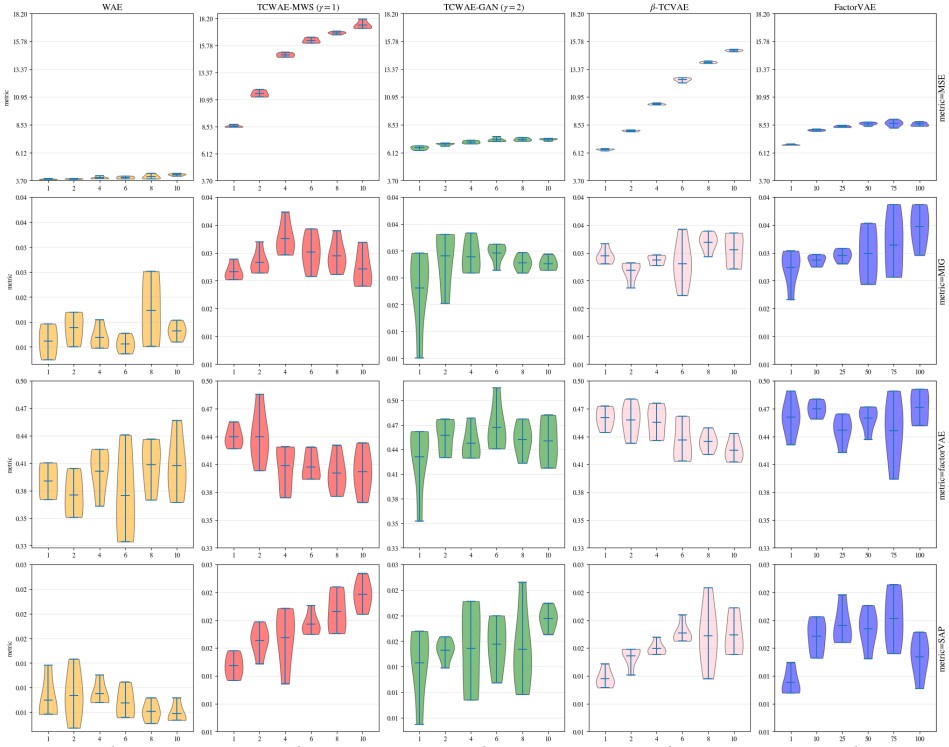

Figure 12: Violin plots of the different scores versus $\gamma$ on smallNORB.

RECONSTRUCTIONS AND SAMPLES

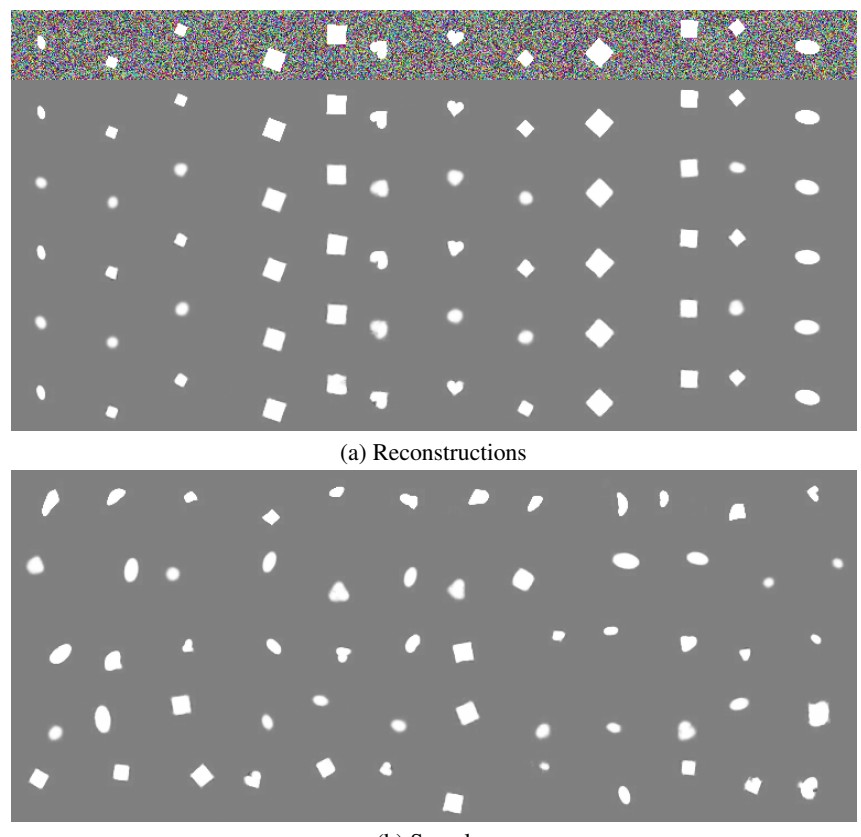

(a) Reconstructions

(b) Samples

Figure 13: Samples and reconstructions for each model on the NoisydSprites. (a): Reconstructions. Top-row: input data, from second-to-top to bottom row: WAE, TCWAE-MWS, TCWAE-GAN , $\beta$-TCVAE, FactorVAE. (b) Samples. From top to bottom row: WAE, TCWAE-MWS, TCWAE-GAN, $\beta$-TCVAE, FactorVAE. Parameters are the ones reported in Tables 1 and 7

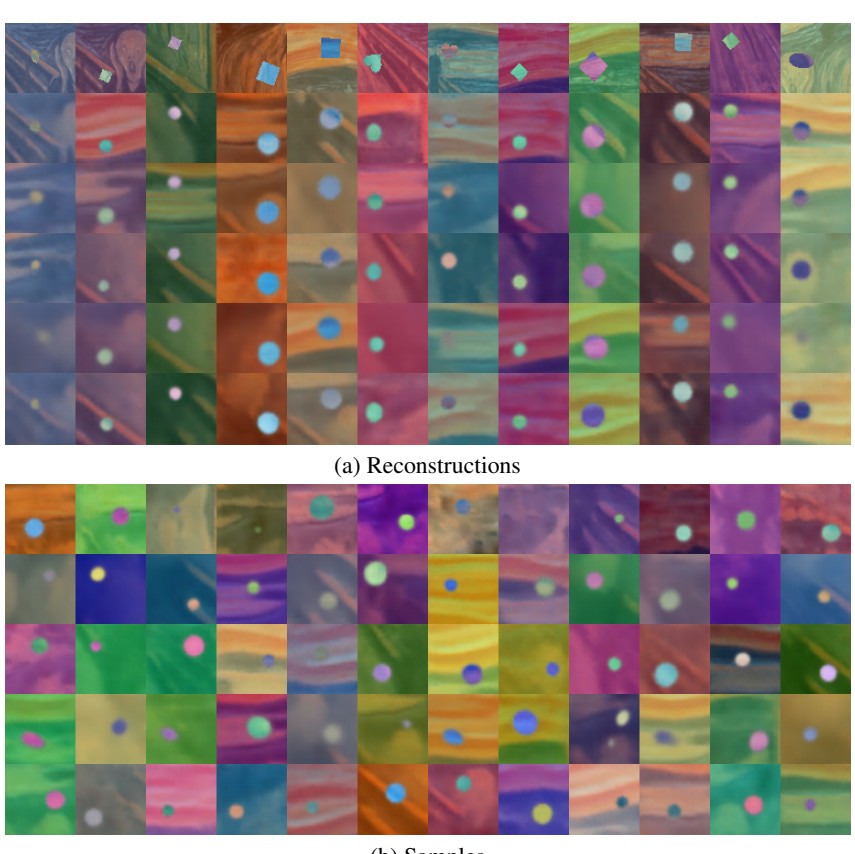

(a) Reconstructions

(b) Samples

Figure 14: Same than Figure 13 but for ScreamdSprites.

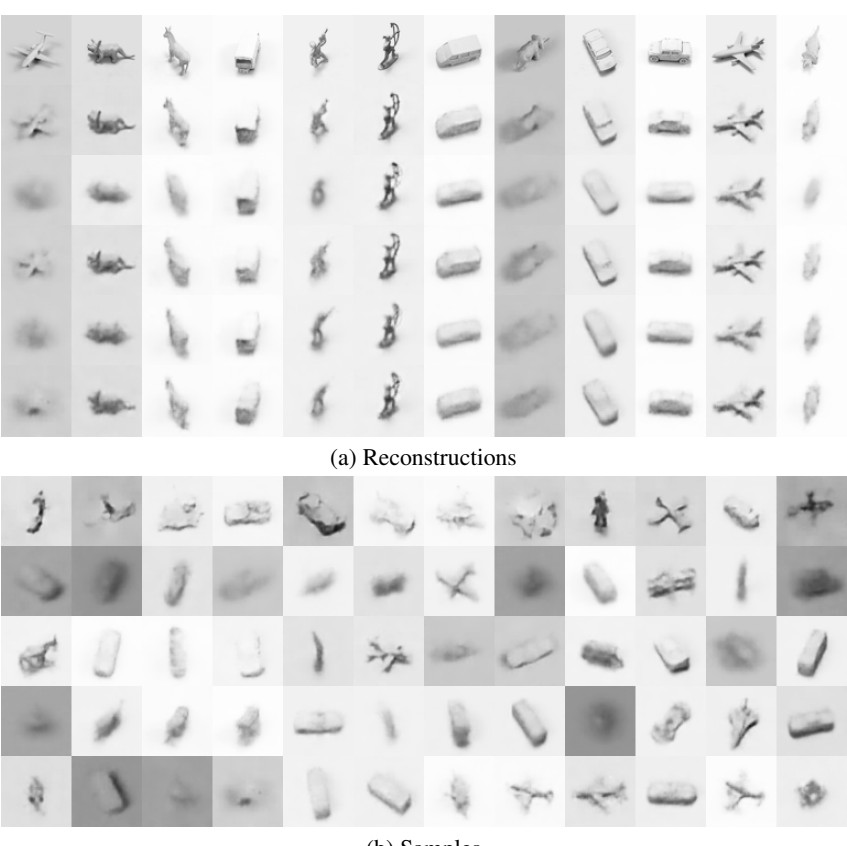

(a) Reconstructions

(b) Samples

Figure 15: Same than Figure 13 but for smallNORB.

# D    QUALITATIVE EXPERIMENTS

3DCHAIRS

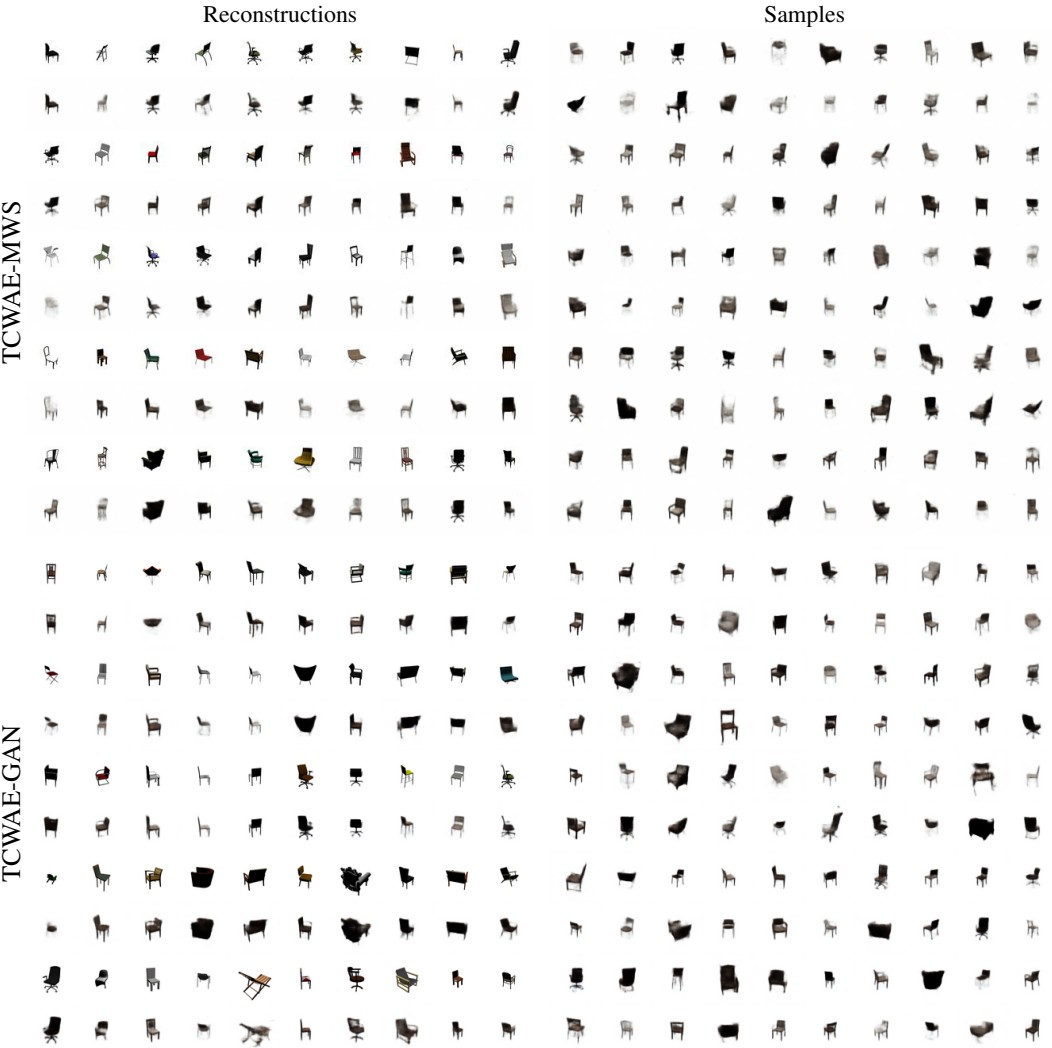

Figure 16: Reconstructions (left quadrants) and samples (right quadrants) for TCWAE-MWS (top quadrants) and TCWAE-GAN (bottom quadrants).

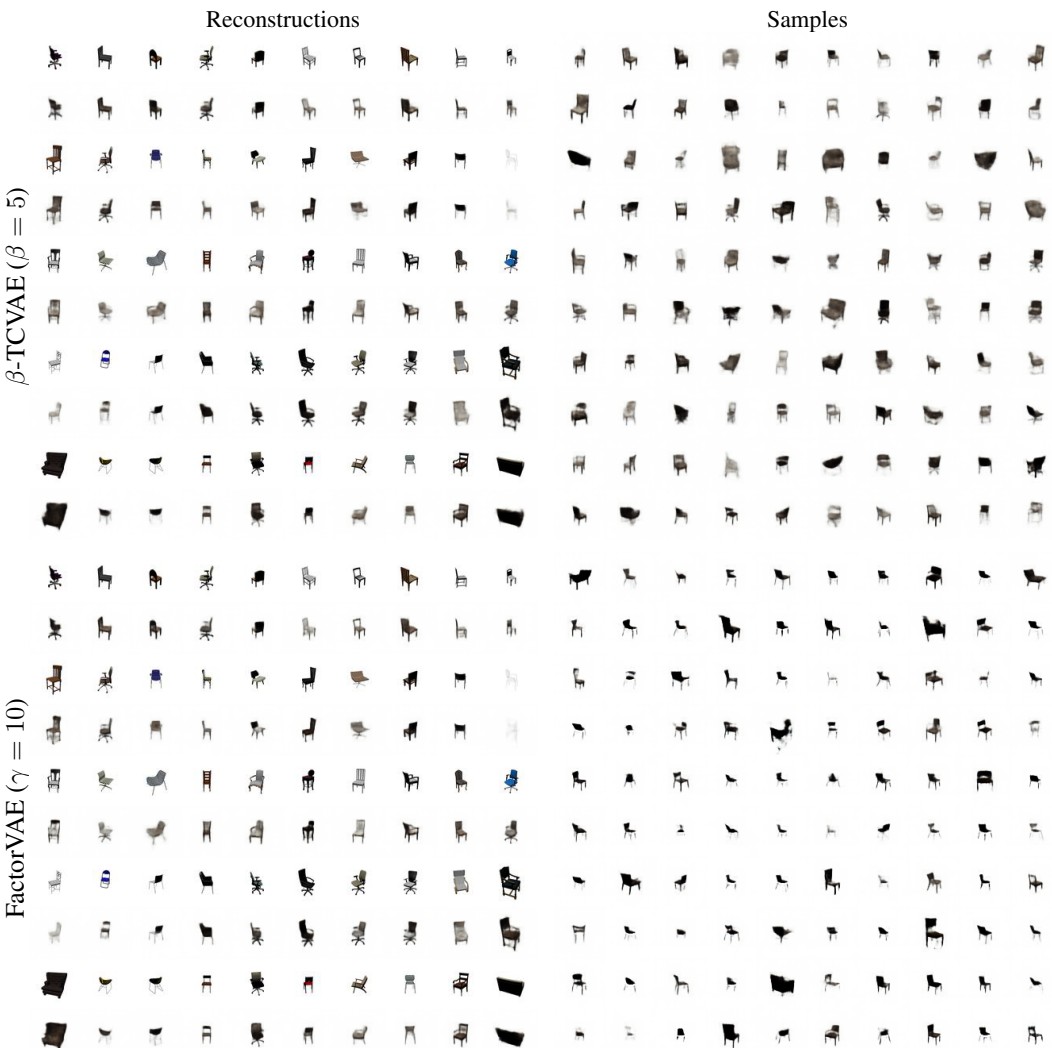

Figure 17: Reconstructions (left quadrants) and samples (right quadrants) for $\beta$-TCVAE (top quadrants) and FactorVAE (bottom quadrants).

CELEBA

Reconstructions                                      Samples

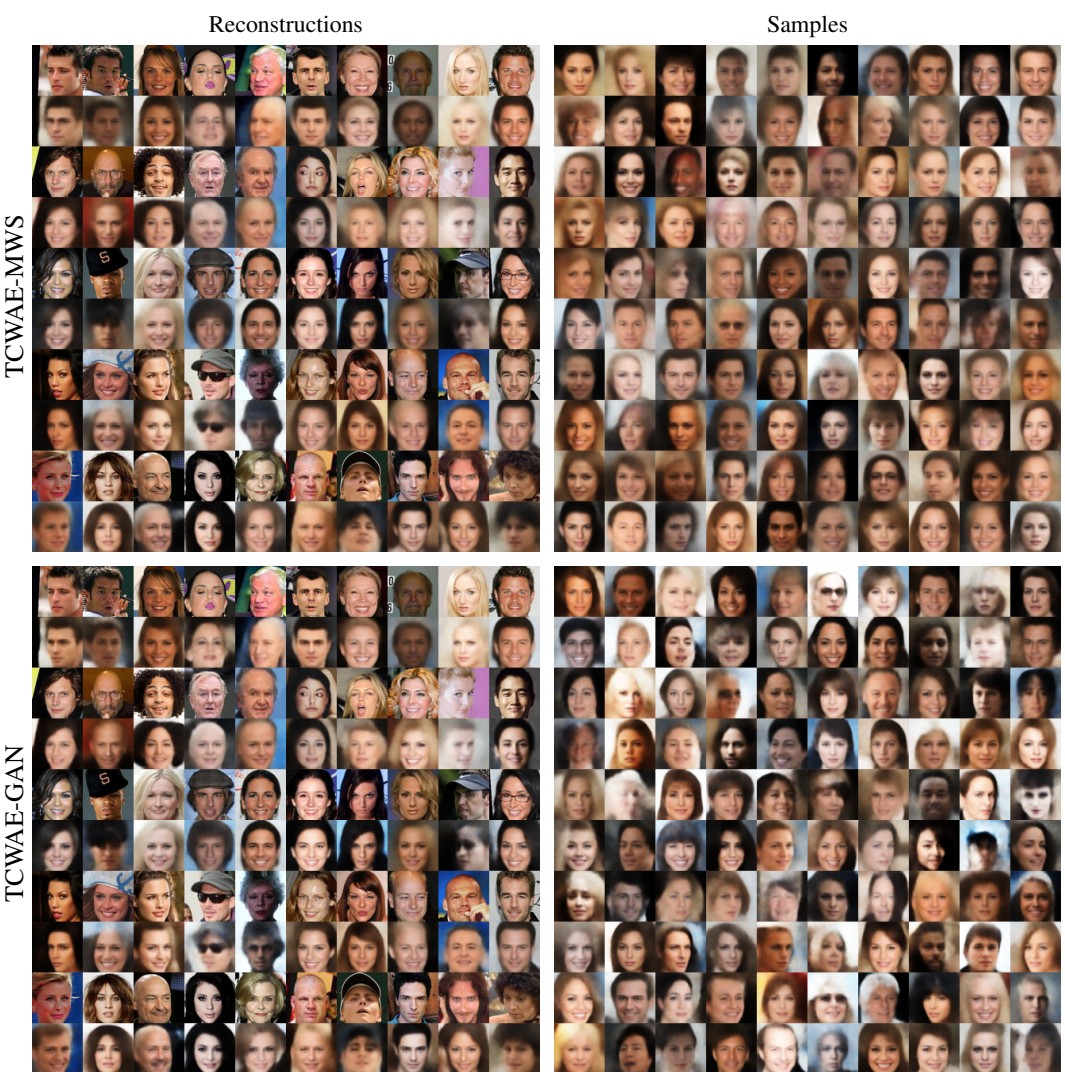

Figure 18: Same as Figure 16 for the CelebA data set.

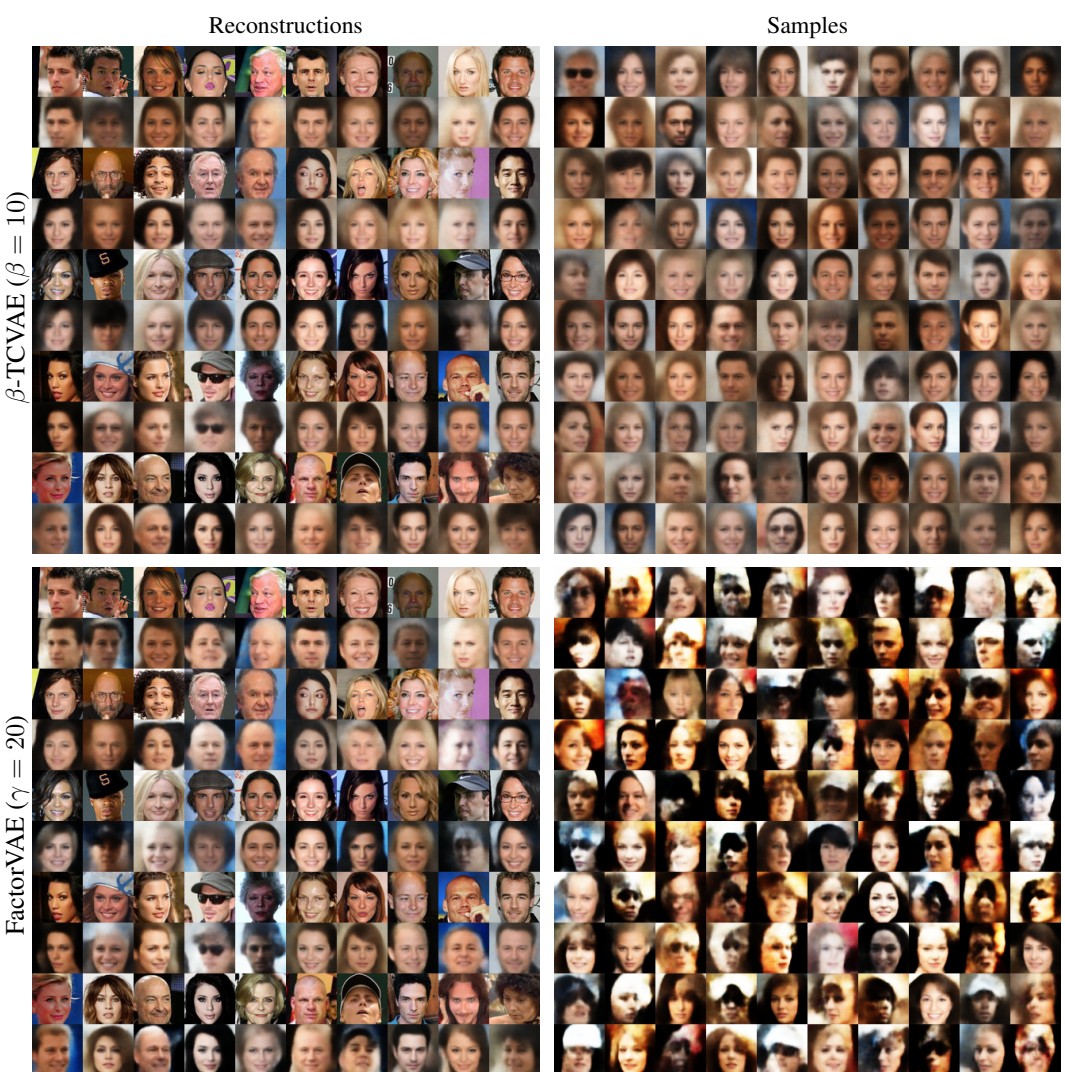

Figure 19: Same as Figure 18 for $\beta$-TCVAE (top quadrants) and FactorVAE (bottom quadrants).

