# OpenReview forum: "Learning disentangled representations with the Wasserstein Autoencoder"
_ICLR.cc/2021/Conference — Reject_

### Official Review · AnonReviewer3 · 2020-10-28
**Official Blind Review**

**Rating:** 8
**Confidence:** 4

**Review:**

Summary:
The paper is motivated by the need for a better trade-off between the reconstruction and disentanglement performance of an autoencoder. The proposed solution is to use KL as a latent regularizer in the framework of Wassestain autoencoders, which allows for a natural interpretation of total correlation.


The paper reads well, all related work and relevant background concepts are nicely integrated throughout the text. The experiments are exhaustive and the results show competitive performance wrt disentanglement while improving reconstruction/modeling of the AEs.

If a dataset is of dynamical nature, how difficult would it be to extend the current version of TCWAE to dynamical systems? Do the authors have any intuition/hint on what should change to make their method applicable to dynamical setups? Significantly changing the probabilistic model or modifying only the and encoder/decoder architecture could suffice?

Minor:
- Consider changing the naming of the baselines either in tables or figures to make them consistent  Chen et al (2018) -> TCVAE Kim & Mnih (2018) -> factorVAE.

---

> ### Author Response · Authors · 2020-11-20
> **Extension to dynamical data**
>
> We thank the reviewers for their comments and will amend the names choices in the figures and tables to make them more consistent.
>
> While static disentanglement learning has been an active centre of interest, disentanglement with dynamic or sequential data has been relatively under studied. Even less so in the WAE framework. It would be indeed interesting to see how to adapt our TCWAE to dynamical data. One objective when working with such data sets is to disentangle the time independent factor (e.g. object class in video, speaker identity in speech) from the dynamical content (e.g. motion in a video, linguistic content in speech). One approach consists in building a generative model with two sets of (independent) latent variables, encoding independently the time dependent information and the the shared and static information respectively [Yingzhen \& Mandt]. How to adapt the TCWAE and especially how to handle the marginals constraints, could be the subject of future works.
>
> Very recently, Recurrent Wasserstein Autoencoder (R-WAE) [Anonymous authors] has been used to learn to disentangle sequential data in the WAE framework. The R-WAE objective possesses two latent regularization terms matching separately the aggregated posteriors of the time-invariant and the time-variant latent to their respective priors. An interesting point is the divergence function chosen by the authors. Indeed, while they propose a version of the R-WAE with the Jensen-Shannon divergence for the time-invariant latent regularization, they argue against a likelihood-based divergence (with min-max optimisation) for the time-variant posterior. This is justified by the fact that the corresponding prior is also dynamically learned, making the min-max optimisation of the divergence harder. It would be interesting to see if a sampled-base estimation of the KL divergence such as the MWS, would be more adapted to such distributions.
>
> **References:**
> * Yingzhen \& Mandt. Disentangled Sequential Autoencoder. International Conference on Machine Learning, 2018.
> * Anonymous authors (under review). Disentangled Recurrent Wasserstein Autoencoder. https://openreview.net/pdf?id=O7ms4LFdsX, 2020

---

### Official Review · AnonReviewer4 · 2020-10-28
**Experimenting with disentanglement, the paper merely reports numbers, but lacks new insight.**

**Rating:** 5
**Confidence:** 4

**Review:**

This paper addresses disentanglement in the latent space of autoencoders. To this end, it combines ideas from four existing papers, namely the reconstruction loss of the Wasserstein autoencoder, the regularization term decomposition from the total correlation autoencoder, and entropy estimation using minibatch-weighted sampling or the density-ratio trick. This combination certainly makes sense, as it brings together methods that have previously been shown to work well in isolation.

The main part of the paper is devoted to an empirical evaluation of the new autoencoder training procedure. The new method is compared against various baselines in terms of L2 reconstruction error and three disentanglement scores on four toy datasets. In addition, latent space traversals on 3Dchairs and CelebA are shown to qualitatively demonstrate the disentanglement capabilities of the proposed methods.

Unfortunately, the description of the experiments is not very precise.
* The role of the hyperparameter gamma remains unclear. In the ablation study, the authors simply set gamma=beta without further explanation, and in the comparison, they just state "we first tune gamma" and "for gamma >1, better disentanglement is obtained", again without further explanation.
* In the comparison experiment, they report results for the values of beta that achieve "an overall best ranking on the four different metrics" without explaining what an "overall best ranking" is. Choices like this must not be taken lightly, as the analysis in "Why rankings of biomedical image analysis competitions should be interpreted with care" (Nature Communications 9: 5217, 2018) impressively demonstrates.
* The experiment in figure 2 seems to have three degrees of freedom (the data instance x, the latent index i, and the size of the modification in direction z_i). However, only two degrees of freedom are shown, and it remains unclear from the caption and associated main text, which ones. Moreover, I cannot deduce justification for the statement "all methods .. learn to disentangle, capturing four different factors" from the figure -- I do not see any obvious disentanglement.

The bigger problem with the paper, however, is the question: What have we learned from these experiments? The rankings in table 1 are pretty inconsistent between different metrics, and the corresponding figure 3 appears to be cherry picked, as the ScreamdSprites is the dataset where the proposed methods perform best.

I also do not agree with the claim that "TCWAEs achieve good disentanglement" on real-world datasets. Figure 4 shows severe entanglement between unrelated factors. For example, the size feature for the chairs also changes the type of chair. All features in the CelebA examples have a tendency also to change the background appearance. The gender feature dramatically influences person identity in the MWS results, whereas it does not change the gender at all in the GAN variant. Substantial variations in person identity are also visible in most other examples.

In summary, while the paper provides numbers, it lacks new insight. In light of mathematical proofs indicating that the true generative factors are generally unidentifiable in non-linear unsupervised settings (cf. the work of Aapo Hyvärinen and others), I am skeptical that heuristic trial-and-error investigations of disentanglement like the present one will yield interesting results. In a sense, this is also acknowledged by the authors, who merely state in the conclusion that "our methods achieve competitive disentanglement on toy data sets" -- that's not much, given the effort that went into the experiments.

---

> ### Author Response · Authors · 2020-11-20
> **Justifying the relevance of our approach**
>
> We thank the reviewer for their feedback.
>
> We will amend the paper so that the experiments and figures, and in general the paper, is easier to read and understand in detail. For example, the latent traversals shown in Figure 2 are obtained after encoding a single observation and traversing each latent dimension at the time, with the traversal range $[-2, 2]$. The reconstruction are then plotted, each row corresponding to a unique latent traversal with the central column corresponding to the true reconstruction.
>
> We also understand the importance of hyper parameters choices and will make sure that the parameters selection process is better described in our work. In particular, $\beta$ is chosen such that the resulting method achieves the best overall score defined as the sum of the ranking on each individual metric:
> $score = r_{MSE} + \sum_{metric} r_{metric}$ where $r_{MSE}$ designed the ranking of the MSE (lower is better) and $r_{metric}$, for $metric$ in \{MIG, FactorVAE, SAP\}, is the ranking of the disentanglement performances as measured by the given metric (higher is better).
>
> In the ablation study, we focused on the impact of using different latent divergence functions in the WAE framework, and thus, by setting the hyper parameters to be the same, we can perform a rigorous comparison of the TCWAE objective with the original WAE. Indeed, when $\beta=\gamma$, TCWAE is simply a WAE with the KL divergence acting as the latent divergence function. Thus, in this setting, we are able to isolate the impact of the latent divergence function on the learning performance of the WAE without any disentanglement performance considerations.
>
> Choosing different values for those regularization weights allows for a fine tuning of the reconstruction-disentanglement trade-off. $\beta$ enforces a low Total Correlation in the aggregated posterior, constraining the aggregated posterior to factorize, resulting in better statistical independence of the latent marginals as discussed in Section 2. $\gamma$ controls the level of marginal-wise regularization to the prior. One would think that its role in disentanglement should be minimal as it mostly prevents the latent codes to drifted away from the prior, resulting in poor generalization and samples generation performances. However, when using a factorized prior, it also implicitly contributes to the factorization of the aggregated posterior (via the the product of its marginal). This is observed in the heat maps Figures 5 to 8 of the Appendix where for several data sets, $\gamma>1$ achieves better disentanglement.
>
> Overall, the impact of the different latent regularization terms does not differ much from the corresponding VAE-based methods, which should not come as a surprise. The only difference in the latent regularization lies in absence of index-code MI term in the TCWAE. However, its role in disentanglement remains to be seen [Chen et al]. Exploring its role in the WAE framework, for example by modifying our objective (Equation (8)) by simply adding an extra regularization term in the form of the index-code MI, could be the subject of future work. However, the resulting objective would only form an upper bound of the original WAE.
>
> Finally, while we acknowledge the not so impressive marginal improvements on toy data sets (Section 4.1), we would argue that this could simply come from to the nature of these data sets (e.g. synthetic, black and white, grey-scale, low resolution), which makes such improvements hard to achieve. On more realistic data sets (Section 4.2) however, the flexibility in the reconstruction term improves the reconstruction and generation performance. While disentanglement remains hard to quantitatively assess when the generative factors are unknown, we would argue that or methods present competitive disentanglement when compared with existing methods.
>
> **References:**
> * Chen et al. Isolating Sources of Disentanglement in VAEs. Advances in Neural Information Processing Systems, 2018.

---

### Official Review · AnonReviewer2 · 2020-11-05
**An well written study, but iterative and without very convincing results**

**Rating:** 5
**Confidence:** 3

**Review:**

This submission proposes to add a KL term to the Wasserstein auto-encoder objective in order to improve its disentanglement capabilities.
This combines the idea of Hoffman & Jonhson (2016) of using a marginal KL term, with the Wasserstein auto-encoder framework.
Challenges regarding the estimation of the KL term are also addressed with two previous works.
This results in a two regularization parameter objective, whose superiority to existing approaches (using a single parameter) is not clear.

Strengths: WAE with a disentanglement term was as far as I now not attempted before, the authors offer two well justified techniques to do it.

Weaknesses: (1) The work is very iterative, existing approaches are only combined, (2) Superiority to WAE without this term is not surprising, and I failed to see a clear superiority to competing unsupervised disentanglement approaches. (3) Given the emphasis on the Wasserstein distance of the original approach, it is also a bit disappointing to resort to a KL term for disentanglement. (4) Most importantly, comparison to simpler alternative KL (non-marginal) losses is absent as far as I can tell. That was for me the most interesting appeal of the paper.

Overall, I tend to think the paper would require a more exhaustive investigation of disentanglement approaches, contextualized to the Wasserstein distance and issues raised regarding marginal versus non-marginal divergences. I recommend rejection.

On this last point: It remains unclear to me whether the original hypothesis of the paper (page 3), that the index-code MI term of the KL divergence may be detrimental to disentanglement, is supported by the current study, and thus whether the extra technicalities required to eliminate it are worth the effort. Perhaps the authors could elaborate on that with an alternative objective close to the classical KL term, and thus easier to optimize?

---

> ### Author Response · Authors · 2020-11-20
> **Novelty of the proposed approach; motivation for the likelihood-based divergence**
>
> We thank the reviewer for their comments and remarks.
> While we did not intend to present our work as a fundamentally new approach, the use of the Wasserstein distance for disentanglement learning has been relatively under studied. Indeed, the use of a KL-based divergence for the latent regularization of the WAE, exploiting its decomposition when using factorized distributions and the role of the resulting terms with respect to information encoding, has never been studied. Thus, in our opinion, while combining existing methods, our work carries a non-negligible level of novelty and relevance.
>
> We recognize that the performance improvements of our method do not represent a step change on toy data sets (Section 4.1). However, we would argue that this can be due to the nature of these data sets (e.g. synthetic, black and white, grey-scale, low resolution), which make such improvements hard to achieve. On more realistic data sets (Section 4.2), we argue that our method achieves better reconstruction and generation performance while retaining visually competitive disentanglement as compared to existing methods. As discussed in the paper, this improvement in the trade-off can be explained by the added flexibility in the reconstruction term of the Wasserstein distance. As rightly suggested by the reviewer, the impact of the index-code MI term on the disentanglement performance remains to be seen. Future work could modify our objective (Equation (8)) to add an extra regularization term in the form of the index-code MI. However, the resulting objective would not provide the same theoretical guaranties than the WAE-based objective, especially in the limit of infinite latent regularization parameter in Equation (8), we would not necessary cover the whole space of possible couplings.
>
> Finally, the use of the KL divergence for latent regularization, motivated by its decomposition and the fine tuning it enables on the level of statistical independence in the latents, is totally legitimate in the WAE framework as both the choice of reconstruction ground-cost and divergence function are at the discretion of the practitioner. Thus, the choice of the KL divergence as a latent regularizer provides the best of both flexible reconstruction cost with a theoretically founded latent regularizer. This is in contrast to the reviewer's comment questioning the justification of using the KL divergence within the Wasserstein framework. On that point, we would like to highlight that the latent divergence function in our objective is indeed the standard KL divergence between the aggregated posterior and the prior. The resulting decomposition in Equation (8) simply unfolds from the use of factorised prior and encoder distribution Q(Z|X). Thus our objective is indeed using the classical KL term but on different distributions than in the VAE-based methods. As mentioned previously, the difference boils down to the presence (in VAE-based methods) or absence (in WAE-based methods) of the index-code MI term. Adding this extra term in our objective would break the WAE approach but could indeed be the object of future work.

---

### Official Review · AnonReviewer5 · 2020-11-05
**Official Blind Review #5**

**Rating:** 6
**Confidence:** 4

**Review:**

This paper extends the Wasserstein Autoencoder (WAE) work by splitting the divergence on the variational marginal into 2 terms, akin to what was done in TC-VAE. This enables directly controlling the explicit contribution of the total correlation term, which is likely to contribute to disentanglement more directly. They explore 2 variations of their model, based on different estimators of the TC term (TCWAE-MWS, using minibatch-weighted sampling; TCWAE-GAN, using a density ratio trick).

Overall, I found this work to be a nicely complete exploration of a simple extension of an existing framework. They mostly rederive existing methods in the WAE framework, but results are promising and the paper addresses several datasets, compares to baselines well and seems well executed. It seems to lack comparison and discussion to a paper which seems directly related [Xiao et al 2019]. But I feel this is still a worthy piece of research to showcase at ICLR.

Questions/comments:
1. A cursory search indicated the following paper which also addresses disentanglement with the Wassertein Total Correlation: [Xiao et al 2019]. They use another estimator of the TC, instead opting for the Kantorovich-Rubinstein formulation.
   1. Can you comment on how their work relates to this current paper?
   2. A direct comparison would be rather interesting, but might be out of scope for a rebuttal.
2. Reconstructions for the TCWAE-MWS appear rather bad (Figures 13-19 in the Appendix ), but Figure 1c doesn’t seem to reflect that, which is slightly surprising.
   1. Could you comment on this discrepancy?
3. Relatedly the TCWAE-GAN disentanglement doesn’t seem particularly exciting (metric-wise), would you still recommend using it instead of TCWAE-MWS?
   1. It is still a clear improvement over vanilla WAE, so there’s value to the work in this current state; but I’d wonder when one would prefer choosing this versus TCVAE?
4. It might be appropriate to discuss 2-Stage VAE [Dai et al 2019] and associated family of models, which currently obtain really good results on more complex datasets.


References:
*	[Xiao et al 2019] https://arxiv.org/abs/1912.12818
*	[Dai et al 2019] https://arxiv.org/abs/1903.05789

---

> ### Author Response · Authors · 2020-11-20
> **Comparison with existing work and clarification of results**
>
> **Comparison to [Xiao et al]:**
>
> We thank the reviewer for rightly directing us to the work of [Xiao et al]. Indeed, it is related to our work in the sense that both methods study the impact on disentanglement of the latent divergence function in the OT framework. While we use the KL divergence, motivated by the works of [Hoffman \& Johnson, Chen et al] and the information theoretic meaning of its decomposition, [Xiao et al] choose instead the 1-Wasserstein distance and, using the triangle inequality, obtain a similar latent regularization decomposition. Thus, the difference boils down to the divergence function: the KL divergence in our case and the 1-Wasserstein distance (in practice WGAN) in [Xiao et al]. [Xiao et al] motivate the use of the 1-Wasserstein distance by the introduction of the Wasserstein total correlation (WTC). In this way, they avoid the difficulty of estimating the KL divergence in a mini-batch setting with intractable distribution (i.e. the aggregated posterior). However, while still a correlation measure, the WTC lacks an information-theoretic interpretation such as the one offered by the TC, namely the loss of information when assuming independent marginals. Moreover, the computation of the 1-Wasserstein distance also comes with its own challenges, especially enforcing the Lipschitz constraint of the critic network remains an open research of area [Arjovsky et al, Gulrajani et al].
> At a higher level, the objective of [Xiao et al] differs from the original WAE, since it resorts to the triangle inequality to obtain the latent regularization decomposition, thus only optimizing an upper bound of the original WAE.
>
> **Clarification of our results and observations:**
>
> When looking at the Figure 1c), the point corresponding to the NoisydSprites reconstruction plots Figure 13a) in the Appendix is the one for which $\beta=\gamma=2$. For those values, the reconstruction performances of TCWAE-MWS lag behind both WAE and TCWAE-GAN. This is partially reflected in the reconstruction plots. However, it is worth remembering that in the ablation study in Figure 1), we fixed $\beta=\gamma$ while for the remaining of the experiments we tuned both $\beta$ and $\gamma$ (especially in the reconstruction plots Figures 13) to 19), $\beta$ and $\gamma$ can be different with the chosen values given in Table 1) and Table 7) respectively.
>
> Globally, we observe that, across all data sets of Section 4.1 bar dSprites, TCWAE-GAN and FactorVAE achieve similar reconstruction performances while TCWAE-MWS performs similarly to TCVAE, with the former methods reconstructing better than the later. This can be seen in Table 1b), where TCWAE-GAN and FactorVAE achieve really close MSE scores, while TCWAE-MWS obtains MSE scores comparable to the ones of TCVAE. This is also reflected in the reconstruction plots. This is not really surprising when looking at the construction of the TCWAE objectives.
>
> As discussed in Section 4.2, TCWAE allow for a more flexible reconstruction cost term as the choice of ground cost function is let to the practitioner. We believe that this added freedom allows our methods to present better reconstruction while retaining the same disentanglement performances than their VAE counterparts. This especially manifests when working with more complex data sets. This is indeed the case in the experiments of Section 4.2, especially with CelebA, where TCWAE show better reconstructions and samples than their VAE counterparts when looking both at the MSE and FID scores. Quantitatively assessing the disentanglement performances remains an open question in the case of unknown generative factors, but the visual inspections of the latent traversals in Figure 4) shows competitive disentanglement performances of our method.
>
> **References:**
> * Xiao et al. https://arxiv.org/abs/1912.12818, 2019.
> * Hoffman \& Johnson. ELBO surgery: yet another way to carve up the variational evidence lower bound. NIPS Workshop on Advances in Approximate Bayesian Inference, 2016.
> * Chen et al. Isolating Sources of Disentanglement in VAEs. Advances in Neural Information Processing Systems, 2018.
> * Arjovsky et al. Wasserstein Generative Adversarial Networks. International Conference on Machine Learning, 2017.
> * Gulrajani et al. Improved Training of Wasserstein GANs. Advances in Neural Information Processing Systems, 2017.

---

### Decision · Program_Chairs · 2021-01-07
**Final Decision**

**Decision:**

Reject

**Comment:**

There were both positive and negative assessments of this paper by the reviewers: It was deemed a well written paper that explores cleanly rederiving the TC-VAE in the Wasserstein Autoencoder Framework and that has experiments comparing to competing approaches. However, there are two strong concerns with this paper: First, novelty appears to be strongly limited as it appears a rederivation using known approaches. Second, two reviewers were not convinced by the experimental results and do not agree with the claim that the proposed approach is better than competing methods in providing disentangled representations. I agree with this concern, in particular as assessing unsupervised disentanglement models is known to be very hard and easily leads to non-informative results (see e.g. the paper cited by the authors  from Locatello et al., 2019). Overall, I recommend rejecting this paper.